# Gender-Specific Malnutrition and Muscle Depletion in Gastric and Colorectal Cancer: Role of Dietary Intake in a Jordanian Cohort

**DOI:** 10.3390/nu16234000

**Published:** 2024-11-22

**Authors:** Nahla Al-Bayyari, Marah Hailat, Ana Baylin

**Affiliations:** 1Department of Nutrition and Food Processing, Faculty of Al-Huson University College, Al-Balqa Applied University, Al-Salt 19117, Jordan; 2Faculty of Medicine, Yarmouk University, Irbid 21163, Jordan; marahhailat97@gmail.com; 3Department of Nutritional Sciences, School of Public Health, University of Michigan, Ann Arbor, MI 48109, USA

**Keywords:** gastrointestinal cancer, malnutrition risk, muscle mass depletion, gastric cancer, colorectal cancer, dietary intake

## Abstract

Objectives: This study aimed to assess malnutrition and muscle mass depletion risk in gastrointestinal cancer patients, exploring the differences between gastric and colorectal cancer, with a focus on gender0specific variations and dietary intake. It also examined whether muscle depletion mediates the relationship between dietary intake and malnutrition risk. Methods: A sample of 100 Jordanian pre-operative gastrointestinal cancer patients (60 male, 40 female) with gastric or colorectal cancer were assessed for malnutrition risk using the malnutrition universal screening tool (MUST) and for muscle depletion using fat-free mass index (FFMI) and mid-upper arm muscle area (MUAMA). Results: The study found that 80% (95% CI: 0.708–0.873) of patients were at high risk of malnutrition, with over 60% experiencing severe muscle loss. Gastric cancer patients showed higher, though not statistically significant, malnutrition risk (90.2% vs. 72.9%) and muscle depletion compared to colorectal cancer patients. Advanced cancer stages were associated with significantly higher risk of malnutrition and muscle depletion. Significant gender-specific differences in muscle depletion via FFMI (*p* = 0.012) and via MUAMA (*p* = 0.028) were also noted, especially in females with gastric cancer. Additionally, males exhibited a significantly higher malnutrition risk (*p* < 0.001) based on cancer stage. Patients’ dietary intake was significantly (*p* < 0.001) below the recommended levels for energy, protein, carbohydrates, fiber, and essential fatty acids, which was associated with higher malnutrition risk, muscle depletion, low BMI (<18.5 kg/m^2^), and significant weight loss (>10%). Low dietary intake was strongly linked to increased malnutrition risk and muscle depletion, with muscle loss partially mediating (b = 0.4972, *p* < 0.0001) the relationship between poor dietary intake and malnutrition risk. Additionally, higher muscle mass was protective against malnutrition (OR = 16.0, 95% CI: 1.706–150.507), and cancer type was a significant predictor of malnutrition risk (OR = 14.4, 95% CI: 1.583–130.867). Conclusions: Malnutrition risk and significant muscle loss are common in GI cancer patients, highlighting the urgent need for tailored nutrition care plans and lifestyle modifications.

## 1. Introduction

Malnutrition is a critical issue in cancer care, affecting between 20% and over 70% of cancer patients globally. Alarmingly, 10% to 20% of cancer-related deaths are attributed to malnutrition rather than the malignancy itself [1]. This issue is particularly pronounced among patients with gastrointestinal (GI) cancers, where malnutrition prevalence can reach up to 80% due to the nature of the disease. The progression of malignancy, the side effects of treatment, and direct tumor effects further exacerbate this condition, leading to significant weight loss and skeletal muscle wasting [2,3].

Given the severe impact of malnutrition on patient outcomes, assessing body composition, as well as skeletal muscle mass specifically, is essential for predicting physical impairment, chemotherapy toxicity, post-operative complications, and mortality in GI cancer patients [4]. Severe muscle mass depletion, often defined as having muscularity below the fifth percentile, can be assessed using techniques such as mid-upper arm area (MUAMA) measurement and bioelectrical impedance to calculate the fat-free mass index (FFMI) [2].

Effective nutrition screening is key to identifying individuals at risk of malnutrition early, ideally before conducting more-detailed assessments [5]. Various screening tools, including the nutrition risk index (NRI), malnutrition universal screening tool (MUST), malnutrition screening tool (MST), and Nutrition Risk Screening 2002 (NRS-2002), have been developed for this purpose [6]. Among these, the MUST has shown the greatest efficacy in identifying malnutrition in elderly patients with GI cancer, as defined by the latest European Society for Clinical Nutrition and Metabolism (ESPEN) diagnostic criteria [7]. Screening for malnutrition with validated risk assessment tools is essential upon hospital admission to quickly identify patients at risk [8]. The MUST has proven to be both accurate and practical for detecting malnutrition risk in hospitalized adults, including GI cancer patients [9].

Research suggests that gender-specific factors can significantly influence the nutritional status and outcomes of cancer patients. Differences in body composition between men and women contribute to varying risks for malnutrition. Men generally have a higher muscle mass, while women tend to have a higher fat mass, which may influence their response to cancer-related malnutrition and treatment [10]. In gastrointestinal cancers, men are more susceptible to muscle mass depletion (sarcopenia), whereas women may experience more fat mass loss; the effects of malnutrition can vary by gender [4]. Additionally, gender can impact dietary intake patterns and metabolic responses. Studies have shown that men and women may have different energy requirements and dietary needs during cancer treatment, affecting their malnutrition risk [11,12,13]. Women may also exhibit distinct dietary behaviors, possibly influenced by sociocultural factors [14].

In males with colorectal cancer and a patient-generated subjective global assessment (PG-SGA) score of ≥9, significant reductions in weight, FFMI, relative fat mass, and serum albumin were observed, while females showed no significant changes but did have elevated C-reactive protein levels. This aligns with lung cancer studies indicating that malnutrition may be more identifiable in males [11,15].

Al-Shahethi et al. [14] reported a higher prevalence of malnutrition in female cancer patients (16.4%) compared to males (15.5%). In contrast, a study by Yi and Hong [16] revealed that male patients with upper gastrointestinal cancer had a greater risk of malnutrition (71%) than females (29%). These differing results suggest that gender-related malnutrition risk may vary depending on the type of cancer and other influencing factors, highlighting the importance of gender-specific strategies in addressing malnutrition.

Inadequate dietary intake is linked to malnutrition risk, with muscle depletion serving as a key indicator, especially in GI cancer patients [16]. This depletion exacerbates frailty, limiting patients’ ability to shop and prepare meals. Moreover, muscle loss may be driven by malnutrition and the inflammatory processes associated with disease states, further worsening the patient’s nutritional status [16,17,18]. These factors combined contribute to a cycle of malnutrition, frailty, and muscle depletion in GI cancer patients, emphasizing the need for early intervention and nutritional support. Therefore, it is expected that muscle depletion might mediate the association between dietary intake and malnutrition risk.

Having recognized the profound influence of nutritional status on clinical outcomes, we aimed to assess the risk of malnutrition and muscle depletion among preoperative GI cancer patients. Furthermore, the study sought to explore potential differences in muscle mass depletion between gastric and colorectal cancer patients, with a particular focus on gender-specific and dietary intake variations and to test if FFMI and/or MUAMA mediated the association between dietary intake and malnutrition risk. The findings of this study could guide targeted nutritional interventions and improve the management of malnutrition in this vulnerable population. We hypothesized that female gastric cancer patients are at a higher risk of malnutrition and muscle depletion, with inadequate dietary intake playing a significant role in their malnutrition.

## 2. Materials and Methods

### 2.1. Study Design and Participants

This study followed a cross-sectional design, enrolling a total of 100 Jordanian adults (60 males and 40 females) diagnosed with either gastric or colorectal cancer. The participants were recruited between January 2018 and March 2020.

Eligibility was determined based on the following criteria: participants were between 40 and 80 years of age, held Jordanian nationality, were attending the GI clinics at King Abdullah University Hospital, had a confirmed diagnosis of gastric or colorectal cancer, had not undergone any GI surgery, could be assessed using the MUST, and had provided signed informed consent confirming that they agreed to participate.

Those meeting the inclusion criteria were selected through a convenience sampling technique.

All necessary medical data, including the time since diagnosis, GI cancer stage based on the TNM classification, and gastric cancer subtype (intestinal or diffuse), were obtained from the patients’ medical records.

Participants were recruited from either the GI surgery clinic or the oncology clinic. Those recruited from the GI surgery clinic were scheduled for surgery, while participants from the oncology clinic were either newly diagnosed, undergoing neoadjuvant therapy, or receiving palliative care.

### 2.2. Ethical Approval

The study was conducted in accordance with the Declaration of Helsinki, following the review and approval of the study protocol by the Institutional Review Board (IRB) at King Abdullah University Hospital (Approval No. 13/111/2018). All participants were fully informed about the study objectives and provided written consent prior to their enrollment.

### 2.3. Anthropometric Measurements and Muscle Depletion

Body weight (kg), fat mass (FM), fat-free mass (FFM), FFMI, total body water (TBW), body mass index (BMI, kg/m^2^), and basal metabolic rate (BMR) were all measured using bioelectrical impedance analysis (BIA) with the Bodystate 1500 MDD Body Composition/Wellness Monitoring Unit. To ensure the accuracy and reliability of the body composition measurements, all preparatory steps were carefully followed. These included standardizing testing conditions, such as ensuring participants were well-hydrated and had refrained from physical activity and caffeine intake prior to testing. This consistency helped minimize variability in the results, providing the most precise and reliable body composition data.

Height was measured using a stadiometer (Seca 700, Hamburg, Germany), with participants barefoot and wearing minimal clothing. They stood with their heels together, arms at their sides, legs straight, shoulders relaxed, and head aligned in the Frankfort horizontal plane [19]. Weight was recorded using a calibrated beam scale, with the patient standing unassisted in the center of the scale, barefoot and in minimal clothing, looking straight ahead [19]. BMI was calculated using Quetelet’s formula: weight (kg) divided by height squared (m^2^) [19]. Waist circumference was measured at the narrowest point between the lowest rib and iliac crest at the end of a normal exhale, while hip circumference was measured at the widest point over light clothing without applying pressure [19].

Triceps skinfold (TSF) thickness was measured using a Holtain skinfold caliper (Holtain, Wales, UK). A trained dietitian pinched a fold of skin and subcutaneous fat about 2 cm above the mid-arm circumference mark. The caliper jaws were positioned over the full skinfold, and the handle was released to apply tension. The thickness was then recorded to the nearest 0.1 mm.

Mid-upper arm circumference (MUAC) was measured using a non-elastic measuring tape (Seca 201, Hamburg, Germany). The measurement was taken on the participant’s right arm, specifically at the midpoint between the acromion (tip of the shoulder) and the olecranon (tip of the elbow). This precise placement ensures an accurate assessment of the arm’s circumference, which is a key indicator of muscle mass and nutritional status. All the measurements were taken to the nearest 0.1 cm (mm). We also calculated the mid-upper arm muscle circumference (MUAMC) using a standard equation [20]: MUAMC (cm) = MUAC (cm) − π × (TSF thickness [millimeters] ÷ 10). From the result of calculation, we calculated the general mid-upper arm muscle area (MUAMA) and the corrected area (CMUAMA) for men and women using the following equations:General equation: MUAMA (cm)^2^ = MUAMC (cm)^2^ ÷ 4 π
For men: CMUAMA (cm)^2^ = ([MUAMC (cm)]^2^ − 10) ÷ 4 π
For women: CMUAMA (cm)^2^ = ([MUAMC (cm)]^2^ − 6.5) ÷ 4 π

Muscle depletion was defined as a CMUAMA below 32 cm^2^ or an FFMI below 14.6 kg/m^2^ for men. For women, muscle depletion was indicated by a CMUAMA below 18 cm^2^ or an FFMI below 11.4 kg/m^2^ [2].

### 2.4. Malnutrition Risk Assessment

The study employed the malnutrition universal screening tool (MUST), a validated, simple, and effective tool widely used for assessing nutritional risk in oncology patients. The MUST is recognized for its applicability across various healthcare settings, including cancer care [21].

All participants were assessed for malnutrition risk using this tool, which evaluates three key factors: body mass index (BMI), unintentional weight loss, and the impact of acute illness, such as taking nothing by mouth for more than five days. Each factor is scored independently on a scale of 0 to 2.

The overall malnutrition risk is classified as low (score = 0), medium (score = 1), or high (score ≥ 2), providing a comprehensive assessment. While each criterion can individually predict clinical outcomes, combining all three offers a more accurate risk prediction [21].

### 2.5. Assessment of Quality of Life

To assess the participants’ quality of life, each individual completed a questionnaire adapted from the EORTC QLQ-C30 [22]. This tool evaluates three key domains: functional abilities, symptoms, and overall health status. It covers a range of topics, including “daily activities, pain, fatigue, appetite, breathing, vomiting, constipation, diarrhea, nausea, sleep, tension, worry, depression, overall health, and quality of life”. The questionnaire was administered after participant enrollment, and scores were calculated based on their responses.

### 2.6. Assessment of Dietary Intake

Participants’ dietary intake was assessed using a three-day food record, provided to each individual upon enrollment in the study. The data collected were analyzed using ESHA software (version 10.6.3) to evaluate adherence to dietary recommendations, which could affect both macro- and micronutrient intake. The results were compared with the recommended dietary allowances (RDAs) for cancer patients. Participants were then categorized into two groups: “adequate” if their intake met the RDAs, and “inadequate” if it fell below the RDAs [23,24]. Nutritional care was considered available when participants adhered to the dietary recommendations provided by the dietitian.

### 2.7. Statistical Analysis

Data were collected, encoded, and analyzed using SPSS version 25. Descriptive statistics, such as means and standard deviations (SDs), were used to summarize continuous variables, while frequencies and percentages described categorical variables. The Kolmogorov–Smirnov test was employed to assess the normality of continuous data.

To compare actual dietary intake with recommended values, a paired-sample *t*-test was applied. Dietary intake was classified as adequate or inadequate based on whether it met the RDA for cancer patients. Quality of life scores were categorized as poor if they were 3 or lower and good if above 3. The MUST score was used to classify malnutrition as high risk for scores ≥ 2, medium risk for a score of 1, and no risk for a score of 0. The time since diagnosis was categorized into two groups: 4 months or less and greater than 4 months, based on the average time since diagnosis among the study participants. Stages of cancer were also categorized into two groups based on the progression of GI cancer: stages I and II and stages III and IV. The Chi-square test then was used to analyze differences in the frequencies of dietary intake, quality of life, and malnutrition risk. For skewed data distributions, the Mann–Whitney U test was used for independent samples, and the Wilcoxon signed-rank test for paired samples. Binary logistic regression was performed to identify the significant predictors for malnutrition risk, while mediation analysis was used to assess the mediation role of FFMI and MUAMA on the relationship between dietary intake and malnutrition risk among the GI cancer patients. A *p*-value of ≤0.05 was considered statistically significant for all tests.

## 3. Results

### 3.1. General Characteristics of the Study Subjects

A total of 856 patients were screened for eligibility to participate in this study. Of these, 756 were excluded, leaving 100 patients (60 males and 40 females) who met the inclusion criteria. The mean age of the participants was 59.2 ± 9.8 years. The average weight was 59.8 ± 8.4 kg, height was 1.7 ± 0.09 m, and the mean BMI was 20.7 ± 2.4 kg/m^2^.

On average, patients experienced 11.9 ± 5.7 kg of weight loss over 4 months. The mean MUST score was 3.1 ± 1.7, indicating a moderate-to-high risk of malnutrition in the cohort. Additionally, the mean waist-to-hip Ratio (WHR) was 0.97 ± 0.1, MUAC was 23.0 ± 2.9 cm, and TSF was 7.3 ± 6.4 mm.

Our sample predominantly comprised the intestinal subtype of adenocarcinoma (*n* = 40), with only one case of the diffuse subtype (*n* = 1). Given this distribution, we opted to group both subtypes under the broader category of gastric cancer in our analyses.

### 3.2. Malnutrition Risk, Muscle Depletion, Adequacy of Dietary Intake, and Quality of Life Scales

Malnutrition risk, muscle depletion, adequacy of dietary intake, and quality of life were assessed based on the type of GI cancer. Table 1 indicates that approximately 63% of the male participants had colorectal cancer. Among the GI cancer patients, 20% had a BMI score below 18.5 kg/m^2^, 85.4% experienced weight loss greater than 10%, and 90% were classified as being at medium-to-high risk of malnutrition.

Overall, 80% (CI: 0.708–0.873) of the study participants were found to be at high risk of malnutrition. However, there were no statistically significant differences (*p* > 0.05) between gastric and colorectal cancer patients in terms of malnutrition risk, BMI, weight loss, acute disease scores, or muscle depletion as measured by both the FFMI and MUAMA. Additionally, no significant differences were observed in the adequacy of dietary intake or in quality of life scores between the two groups.

Despite this, 60% of the patients reported poor functional status and symptom scale scores, indicating a diminished quality of life.

A comparison of GI cancer patients revealed no statistically significant differences between male and female patients in terms of BMI (*p* = 0.495) or weight loss (*p* = 0.157). However, a significant difference was found in their acute disease scores (*p* = 0.007), with 43.3% of males receiving a score of two.

Additionally, no statistically significant differences were observed regarding malnutrition risk (*p* = 0.203), healthcare scale (*p* = 0.824), symptom scale (*p* = 0.082), adequacy of dietary intake (*p* = 0.298), or the type of GI cancer (*p* = 0.539) between males and females.

On the other hand, significant differences were detected in the functional scale (*p* = 0.006), where 75% of males had poor functional scale scores. Muscle depletion indicators also showed significant differences between the sexes. For the FFMI, 75% of males had depleted muscle mass compared to 50% of females (*p* = 0.018), and for MUAMA, 63% of males had muscle depletion compared to 32.5% of females (*p* = 0.001).

Table 2 presents the malnutrition risk, muscle depletion, dietary intake adequacy, and quality of life scales according to both the type of GI cancer and sex. The results showed no statistically significant differences between male gastric and colorectal cancer patients across these variables.

However, among female patients, significant differences (*p* < 0.05) were found in BMI, weight loss scores, malnutrition risk, muscle depletion as measured by FFMI and MUAMA, and adequacy of dietary intake. Female patients with gastric cancer exhibited higher frequencies of these adverse outcomes compared to those with colorectal cancer.

### 3.3. Malnutrition Risk, Muscle Depletion, and Quality of Life Scales According to Cancer Stage and Time Since Diagnosis

Table 3 demonstrates statistically significant differences (*p* < 0.05) in BMI scores, weight loss scores, acute disease scores, malnutrition risk, muscle depletion (assessed by FFMI and MUAMA), dietary intake, and functional and symptom-related quality of life scales according to cancer stages. A higher percentage of patients with GI cancer in stages I and II had BMI scores above 20 kg/m^2^ compared to those in stages III and IV. Conversely, among patients in stages III and IV, more patients experienced weight loss exceeding 10%, and they showed a greater risk of malnutrition, more muscle depletion, poorer functional and symptom-related quality of life scores, and more frequent instances of inadequate dietary intake.

When analyzing differences in GI cancer stages based on gender, results revealed no significant differences in malnutrition risk, muscle depletion, dietary intake, or any quality of life scales among females. However, male patients showed significant differences (*p* < 0.05) in weight loss scores, acute disease scores, and malnutrition risk according to cancer stage. Among males, more patients in stages III and IV experienced weight loss exceeding 10%, had acute disease scores of two, and showed higher malnutrition risk compared to those in stages I and II (Table 3).

Additionally, no significant differences were observed in malnutrition risk, muscle depletion, dietary intake, or any quality of life scales when the data were compared based on the time since diagnosis. However, we found a significantly higher proportion of males with a time since diagnosis greater than 4 months (Table 3).

### 3.4. Dietary Intake

Macronutrient intake was compared to the recommended dietary allowances (RDAs) for all cancer patients. The results presented in Table 4 revealed statistically significant differences between the mean ± SD of actual dietary intake and the RDAs for total energy, protein, carbohydrates, fiber, and omega-3 and omega-6 fatty acids. The intake of these macronutrients among patients was significantly lower than the recommended levels.

Based on the above results, we compared malnutrition risk, muscle depletion, quality of life scales, type of GI cancer, cancer stage, and time since diagnosis according to the adequacy of dietary intake and sex. The results presented in Table 5 show significant differences (*p* < 0.05) in all variables except for sex, type of GI cancer, healthcare score, and time since diagnosis.

When comparing these variables based on dietary intake adequacy and sex, significant differences were found in BMI scores, weight loss scores, malnutrition risk, muscle mass depletion (measured by FFMI and MUAMA), and symptom scales. These adverse outcomes were observed to be more common among both male and female patients with inadequate dietary intake.

In addition, significant differences were detected in acute disease score and functional scale among male patients but not among female patients. Specifically, male patients with inadequate dietary intake were more likely to receive a score of two on the acute disease scale and poor functional scale outcomes. Neither cancer stage nor the time of diagnosis showed significant differences in dietary intake between male and female patients.

### 3.5. Mediation Analysis

This study examined the mediating effect of FFMI and MUAMA as indicators of muscle mass depletion on the relationship between dietary intake and malnutrition risk in patients with GI cancer. The findings revealed a significant indirect effect of dietary intake on malnutrition risk via FFMI (b = 0.0285, t = 0.5852). Additionally, a significant indirect effect was observed through MUAMA (b = 0.0878, t = 1.5791).

The direct effect of dietary intake on malnutrition risk, accounting for the mediators (FFMI and MUAMA), was also significant (b = 0.2849, *p* = 0.0091), indicating partial mediation by both FFMI and MUAMA. Furthermore, sex was found to be a significant covariate, influencing both FFMI (b = −1.4618, t = −3.9862, *p* = 0.0001) and MUAMA (b = −8.1623, t = −7.4218, *p* < 0.0001), although its effect on malnutrition risk was not significant (b = 0.1036, t = 0.9883, *p* = 0.3255). Meanwhile, the cancer stage covariate did not have a significant effect on either the FFMI (b = −0.6795, t = −1.5834, *p* = 0.01166) or the MUAMA (b = 0.9087, t = 0.7060, *p* = 0.4819). However, it had a highly significant effect on malnutrition risk (b = 0.6378, t = 6.6145, *p* < 0.0001) A summary of the mediation analysis is provided in Table 6 and Figure 1.

Dietary intake had a significant effect on FFMI (b = −2.1718, t = −5.2342, *p* < 0.0001), MUAMA (b = 2.6049, t = 2.0934, *p* = 0.0039), and malnutrition risk (b = 0.2849, t = 2.6619, *p* = 0.0091). While FFMI did not show a significant effect on malnutrition risk (b = −0.0131, t = −0.5772, *p* = 0.5652), MUAMA significantly impacted malnutrition risk (b = 0.0337, t = 4.4495, *p* < 0.0001). The total effect of dietary intake on malnutrition risk was also significant (b = 0.4012, t = 4.0203, *p* = 0.0001) (Figure 1).

### 3.6. Predictors of Malnutrition Risk

Table 7 presents the results of the logistic regression model, identifying significant predictors for malnutrition risk. Four key variables—sex, energy intake, MUAMA, and the type of GI cancer—showed a statistically significant contribution to the model.

The analysis indicates that females are 14.5 times more likely to be at risk of malnutrition compared to males, after adjusting for the other variables in the model. Energy intake is shown to be a protective factor, as higher daily energy intake is associated with a decreased risk of malnutrition. Additionally, for each unit increase in MUAMA, the odds of malnutrition decrease substantially, with a reduction in risk by approximately 16-fold compared to individuals with normal MUAMA. This suggests that greater muscle mass is a strong protective factor against malnutrition. Furthermore, the type of cancer also plays a critical role in predicting malnutrition risk. Patients diagnosed with gastric cancer are 14.4 times more likely to experience malnutrition compared to those with colorectal cancer. This highlights the higher vulnerability of gastric cancer patients to malnutrition, possibly due to the nature of the disease and its impact on nutrient absorption and digestive function.

In summary, sex, energy intake, MUAMA, and the type of GI cancer are crucial determinants of malnutrition risk. Being female, having lower energy intake and MUAMA, and being diagnosed with gastric cancer significantly increase the likelihood of malnutrition.

## 4. Discussion

This study highlights the high prevalence of malnutrition in gastric and colorectal cancer patients, with gastric cancer patients at a higher risk of malnutrition and muscle depletion, especially females. Patients’ intake of essential nutrients, including energy and protein, was significantly below recommended levels, worsening malnutrition risk. Inadequate intake was linked to greater muscle loss, lower BMI, and severe weight loss. Muscle depletion partially mediated the relationship between diet and malnutrition risk. Greater muscle mass and energy intake protected against malnutrition, with cancer type and gender influencing risk. The findings emphasize the importance of nutritional interventions for cancer patients.

Malnutrition affects 20–70% of cancer patients globally, with GI cancer patients being particularly at risk [1]. In GI cancers, malnutrition prevalence can reach up to 80–90% [2,25], driven by disease progression, treatment side effects, and the impact of the tumor on digestion and metabolism, leading to weight loss and muscle wasting [2,3]. This compromises patients’ ability to handle surgical stress and affects postoperative recovery [26]. Factors like inadequate food intake, chronic illness, metabolic changes, psychological stress, and low socioeconomic status contribute to preoperative malnutrition, increasing complications, mortality rates, hospital stays, and healthcare costs [12,13,27].

A recent multicenter study in Spain assessed 469 gastrointestinal cancer patients (85% aged > 40 years) for malnutrition risk using the MUST and the Global Leadership Initiative on Malnutrition (GLIM) criteria. It was found that 17.9% were at moderate risk (MUST score of one) and 21.1% were at high risk (MUST score of two or above). Among high-risk patients, 35.3% were moderately and 64.6% severely malnourished according to the GLIM [18]. Yi and Hong [16] used the subjective global assessment (SGA) tool on 409 upper GI cancer patients in Malaysia, and found 92.1% to be malnourished. Our study aligns with these findings, showing that 80% of GI cancer patients are at high risk, with 90.2% of gastric and 73.9% of colorectal cancer patients affected. However, our results disagree with the findings of Durán et al. [18].

The high prevalence of malnutrition in GI cancer patients is linked to symptoms like poor appetite, nausea, vomiting, dysphagia, dyspepsia, and diarrhea, which affect food intake and nutrient absorption [25]. While this study did not collect detailed data on symptom severity, the symptom scale used revealed a significant association between worsening symptoms and inadequate dietary intake. This suggests that severe symptoms may worsen nutritional intake and increase malnutrition risk. Though lacking precise symptom severity data, the symptom scale offers valuable insights into the relationship between symptoms and malnutrition among GI cancer patients.

Despite the high prevalence of malnutrition in cancer patients, nutritional assessments are not routinely conducted [17,28], and 50% of malnourished patients remain undiagnosed or untreated [29].

Yi and Hong [16] used the Eastern Cooperative Oncology Group scale (ECOG), which is commonly used to assess physical function in oncology [26], and found that 61% of upper GI cancer patients had impaired functional status (ECOG scores of 2–3), linked to muscle loss, poor food intake, and depression. In our study, 64% of GI cancer patients showed poor functional status (EORTC QLQ-C30 scale) [22], with 78.3% having inadequate dietary intake, likely contributing to their functional decline.

Preoperative serum albumin is a useful marker for detecting malnutrition and predicting outcomes, with early intervention in hypoalbuminemia potentially reducing complications [30]. However, in our study, malnutrition risk was assessed using BMI, weight loss, and MUST scores, while muscle depletion was measured by FFMI and MUAMA. Biochemical markers like serum albumin were not included alongside these anthropometric measures and body composition tools, though they could have offered a more comprehensive evaluation of malnutrition risk.

Previous studies have shown that malnutrition is more common at advanced cancer stages [16], primarily due to factors such as reduced appetite, changes in taste and smell, nausea, difficulties with eating (e.g., chewing or dysphagia), vomiting, impaired nutrient absorption, and systemic inflammation [31]. Consistent with these findings, our study also identified a higher risk of malnutrition among patients with advanced stages of GI cancer. Additionally, we observed significant gender differences in malnutrition risk, with males showing a higher overall malnutrition risk in relation to cancer stage. These findings align with previous research that highlighted gender-based disparities in malnutrition prevalence among cancer patients [11,14].

A recent study suggests men present more-consistent malnutrition symptoms, making diagnosis easier [11]. However, our study found higher malnutrition risk among men (85%) than women (72.5%). Females with gastric cancer showed a significantly higher risk of malnutrition and muscle depletion compared to those with colorectal cancer. While overall dietary intake showed no sex-based differences, females with gastric cancer showed a higher prevalence of inadequate intake. Among males, no significant intake differences were seen between cancer types. Despite the easier identification of malnutrition in men, women were more likely to be malnourished, likely due to lower fat and muscle mass, as indicated by bioelectrical impedance analysis.

Two studies reported a low muscle mass prevalence of 15% to 19.2% in newly diagnosed cancer patients using FFMI estimated from BIA [32,33], which is lower than that found in our study, likely due to differences in cancer type. BIA’s limitations in assessing FFM, especially in metastatic patients, may contribute to inaccuracies [34]. Malnutrition rates of up to 77% have been reported in advanced cancer patients using the GLIM criteria and FFMI [35].

In the UK Biobank cancer cohort, malnutrition was more common than low muscle mass and sarcopenia, all linked to higher mortality [36]. Similarly, our study found higher malnutrition risk (80%) compared to muscle depletion (60–65%), with muscle depletion partially mediating the relationship between dietary intake and malnutrition.

A study on colorectal cancer patients found no significant effect of visceral adipose tissue (VAT) or subcutaneous adipose tissue (SAT), attained from CT scans, on survival, but higher muscle mass improved 5-year survival rates, influenced by age, cancer stage, and surgery [37]. Although we did not assess survival, our analysis showed that greater muscle mass protected against malnutrition, which is linked to lower survival.

Previous studies have shown that gastric cancer patients are more prone to malnutrition compared to those with colorectal cancer. This increased risk is often due to insufficient nutrient intake or absorption, leading to significant weight loss, alterations in body composition, and worse clinical outcomes [38]. In line with these findings, our study revealed that 85.4% of gastric cancer patients experienced weight loss greater than 10%, and 68.3% had inadequate dietary intake. In comparison, colorectal cancer patients showed lower rates of both weight loss (69.5%) and inadequate intake (54.2%). This weight loss is likely a result of reduced food consumption, compounded by GI symptoms [27].

A comprehensive dietary assessment, including current and past intake, should be conducted alongside the use of tools like the SGA. Yi and Hong [16] found that GI cancer patients consumed less than 25 kcal/kg/day. Our study also showed energy and protein intake to be below the recommended levels. Malnutrition in GI cancer patients is often driven by mechanical obstructions, cancer cachexia, and anorexia [17,39].

The ESPEN highlights the need for early screening and the continuous monitoring of nutrition, muscle mass, and inflammation in cancer patients [28]. Once malnutrition risk is identified, assessment by a dietitian and multidisciplinary care are essential [12,17]. Preoperative nutritional interventions can improve patient outcomes by enhancing nutritional status and recovery after surgery [40].

This study supports our initial hypothesis but has limitations, including a small sample size, cross-sectional design, and reliance on BIA and anthropometric measurements instead of more-accurate CT scans for muscle depletion. The absence of biochemical markers such as serum albumin further limit the findings’ accuracy and generalizability.

## 5. Conclusions

In conclusion, this study reveals a high prevalence of malnutrition and muscle depletion in gastrointestinal cancer patients, particularly in women with gastric cancer and men with advanced-stage GI cancer. Poor nutrient intake worsened these risks, leading to weight loss and reduced BMI. Muscle depletion partially mediated the link between diet and malnutrition. Routine nutritional screening and early interventions are vital to improving outcomes and survival rates.

## Figures and Tables

**Figure 1 nutrients-16-04000-f001:**
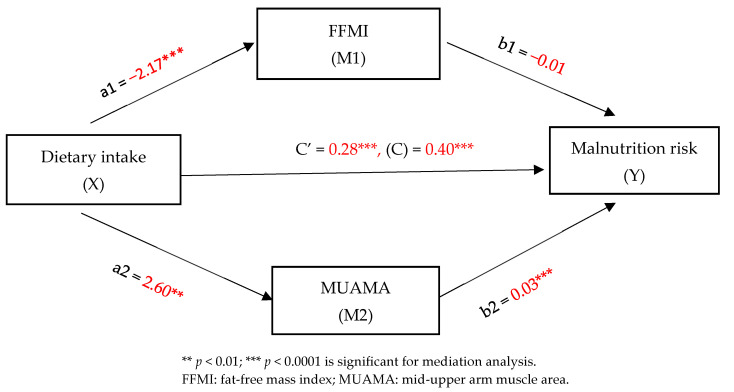
Regression coefficients from the mediation analysis using Hayes Process Macro Model 4 with two mediators and with age and cancer stage as covariates.

**Table 1 nutrients-16-04000-t001:** Malnutrition risk, muscle depletion, adequacy of dietary intake, quality of life scales, cancer stage, and time since diagnosis according to the type of gastrointestinal cancer (N = 100).

Variable	Type of Cancer			*p*-Value
	All (N = 100)	Gastric (n = 41)	Colorectal (n = 59)	
	n (%)	n (%)	n (%)	
Sex				
- Male - Female	60 (60.0)	23 (56.1)	37 (62.7)	0.323
	40 (40.0)	18 (43.9)	22 (37.3)	
BMI score (kg/m^2^)				
- >20 - 18.5–20 - <18.5	51 (51.0)	17 (41.5)	34 (57.6)	0.216
	35 (35.0)	16 (39.0)	19 (32.2)	
	14 (14.0)	8 (19.5)	6 (10.2)	
Weight loss score (%)				
- <5 - 5–10 - >10	9 (9.0)	1 (2.4)	8 (13.6)	0.103
	15 (15.0)	5 (12.2)	10 (16.9)	
	76 (76.0)	35 (85.4)	41 (69.5)	
Acute disease score				
- Zero - One - Two	46 (46.0)	17 (41.5)	29 (49.2)	0.659
	20 (20.0)	8 (19.5)	12 (20.3)	
	34 (34.0)	16 (39.0)	18 (30.5)	
Malnutrition risk				
- No risk - Medium risk - High risk	9 (9.0)	1 (2.4)	8 (13.6)	0.093
	11 (11.0)	3 (7.3)	8 (13.6)	
	80 (80.0)	37 (90.2)	43 (72.9)	
Muscle depletion according to FFMI (kg/m^2^)				
- Normal - Depleted	35 (35.0)	12 (29.3)	23 (39.0)	0.216
	65 (65.0)	29 (70.7)	36 (61.0)	
MUAMA (cm^2^)				
- Normal - Depleted	40 (40.0)	14 (34.1)	26 (44.1)	0.397
	60 (60.0)	27 (65.9)	33 (55.9)	
Dietary intake				
- Adequate - Inadequate	40 (40.0)	13 (31.7)	27 (45.8)	0.114
	60 (60.0)	28 (68.3)	32 (54.2)	
Healthcare scale				
- Good - Bad	70 (70.0)	27 (65.9)	43 (72.9)	0.296
	30 (30.0)	14 (34.1)	16 (27.1)	
Functional scale				
- Good - Bad	36 (36.0)	16 (39.0)	20 (33.9)	0.376
	64 (64.0)	25 (61.0)	39 (66.1)	
Symptoms scale				
- Good - Bad	32 (32.0)	12 (29.3)	20 (33.9)	0.395
	68 (68.0)	29 (70.7)	39 (66.1)	
Cancer stage				
- I and II - III and IV	33 (33.0)	9 (22.0)	24 (40.7)	0.056
	67 (67.0)	32 (78.0)	35 (59.3)	
Time since diagnosis				
- ≤4 months - >4 months	68 (68.0)	27 (65.9)	41 (69.5)	0.825
	32 (32.0)	14 (34.1)	18 (30.5)	

Data are presented as frequencies (n) and percentages (%). BMI: body mass index; FFMI: fat-free mass index; MUAMA: mid-upper arm muscle area. Muscle depletion was defined as FFMI < 14.6 kg/m^2^ or MUAMA < 32 cm^2^ for males and FFMI < 11.4 kg/m^2^ or MUAMA < 18 cm^2^ for females. *p* < 0.05 was considered significant using the Chi-square test or Fisher’s exact test (applied when cell counts were <5).

**Table 2 nutrients-16-04000-t002:** Malnutrition risk, muscle depletion, adequacy of dietary intake, quality of life scales, cancer stage, and time since diagnosis according to the type of gastrointestinal cancer and sex (N = 100).

Variable	Male (n = 60)		*p*-Value	Female (n = 40)		*p*-Value
	Type of Cancer			Type of Cancer		
	Gastric (n = 23) n (%)	Colorectal (n = 37) n (%)		Gastric (n = 18) n (%)	Colorectal (n = 22) n (%)	
BMI score (kg/m^2^)						
- >20 - 18.5–20 - <18.5	11 (47.8)	17 (45.9)	1.000	6 (33.3)	17 (77.3)	0.004
	8 (34.8)	14 (37.8)		8 (44.4)	5 (22.7)	
	4 (17.4)	6 (16.2)		4 (22.3)	0.5 (0.0) ^	
Weight loss score (%)						
- <5 - 5–10 - >10	1 (4.3)	2 (5.4)	1.000	^ 0.5 (0.0)	6 (27.3)	0.016
	3 (13.0)	5 (13.5)		2 (11.1)	5 (22.7)	
	19 (82.6)	30 (81.1)		16 (88.9)	11 (50.0)	
Acute disease score						
- Zero - One - Two	7 (30.4)	13 (35.1)	0.524	10 (55.6)	16 (72.7)	0.424
	4 (17.4)	10 (27.0)		4 (22.4)	2 (22.4)	
	12 (52.2)	14 (37.8)		4 (22.4)	4 (18.2)	
Malnutrition risk						
- No risk - Medium risk - High risk	1 (4.3)	2 (5.4)	0.633	^ 0.5 (0.0)	6 (27.3)	0.039
	1 (4.3)	5 (13.5)		2 (11.1)	3 (13.6)	
	21 (91.4)	30 (81.1)		16 (88.9)	13 (59.1)	
Muscle depletion according to FFMI (kg/m^2^)						
- Normal - Depleted	7 (30.4)	8 (21.6)	0.320	5 (27.8)	15 (68.2)	0.012
	16 (59.6)	29 (78.4)		13 (72.2)	7 (31.8)	
Muscle depletion according to MUAMA (cm^2^)						
- Normal - Depleted	8 (43.8)	11 (29.7)	0.448	6 (33.3)	15 (68.2)	0.028
	15 (65.2)	26 (70.3)		12 (66.7)	7 (31.8)	
Dietary intake						
- Adequate - Inadequate	8 (34.8)	13 (35.1)	0.601	5 (27.8)	14 (63.6)	0.025
	15 (65.2)	24 (64.9)		13 (72.2)	8 (36.4)	
Healthcare scale						
- Good - Bad	15 (65.2)	26 (70.3)	0.778	12 (66.7)	17 (77.3)	0.347
	8 (34.8)	11 (29.7)		6 (33.3)	5 (22.7)	
Functional scale						
- Good - Bad	7 (30.4)	8 (21.6)	0.320	9 (50.0)	12 (54.2)	0.512
	16 (69.6)	29 (78.4)		9 (50.0)	10 (45.5)	
Symptoms scale						
- Good - Bad	6 (26.1)	9 (24.3)	0.556	6 (33.3)	11 (50.0)	0.230
	17 (73.9)	28 (75.7)		12 (66.7)	11 (50.0)	
Cancer stage						
- I and II - III and IV	6 (26.1)	13 (35.1)	0.573	5 (27.8)	9 (40.9)	0.510
	17 (73.9)	24 (64.9)		13 (72.2)	13 (59.1)	
Time since diagnosis						
- ≤4 months - >4 months	12 (52.2)	23 (62.2)	0.446	15 (83.3)	18 (81.8)	1.000
	11 (47.8)	14 (34.8)		3 (16.7)	4 (18.2)	

Data are presented as frequencies (n) and percentages (%). BMI: body mass index; FFMI: fat-free mass index; MUAMA: mid-upper arm muscle area. Muscle depletion was defined as FFMI < 14.6 kg/m^2^ or MUAMA < 32 cm^2^ for males and FFMI < 11.4 kg/m^2^ or MUAMA < 18 cm^2^ for females. *p* < 0.05 was considered significant using the Chi-square test or Fisher’s exact test (applied when cell counts were <5). ^ Fisher’s exact test assumes cells with 0 are approximately 0.5.

**Table 3 nutrients-16-04000-t003:** Malnutrition risk, muscle depletion, and quality of life scales according to cancer stage, sex, and time since diagnosis of the gastrointestinal cancer patients (N = 100).

Variable	Cancer Stage			Time Since Diagnosis			Male Cancer Stage			Female Cancer Stage		
	I and II (n = 33)	III and IV (n = 67)	*p*	≤4 Months (n = 68)	>4 Months (n = 32)	*p*	I and II (n = 19)	III and IV (n = 41)	*p*	I and II (n = 14)	III and IV (n = 26)	*p*
	n (%)	n (%)		n (%)	n (%)		n (%)	n (%)		n (%)	n (%)	
Sex												
- Male - Female	19 (57.6)	41 (61.2)	0.829	35 (51.5)	25 (78.1)	0.016	NA	NA	NA	NA	NA	NA
	14 (42.4)	26 (38.8)		33 (48.5)	7 (21.9)							
Type of cancer												
- Gastric - Colorectal	9 (27.3)	32 (47.8)	0.056	27 (39.7)	14 (43.8)	0.828	6 (31.6)	17 (41.5)	0.573	5 (35.7)	13 (50.0)	0.510
	24 (72.7)	35 (52.2)		41 (60.3)	18 (56.3)		13 (68.4)	24 (58.5)		9 (64.3)	13 (50.0)	
BMI score (kg/m^2^)												
- >20 - 18.5–20 - <18.5	25 (75.8)	26 (38.8)		35 (51.5)	16 (50.0)		13 (68.4)	15 (36.6)		8 (57.1)	15 (57.7)	
	7 (21.2)	28 (41.8)	0.001	26 (38.2)	9 (28.1)	0.245	4 (21.1)	18 (43.9)	0.092	6 (42.9)	7 (26.9)	0.339
	1 (3.0)	13 (19.4)		7 (10.3)	7 (21.9)		2 (10.5)	8 (19.5)		^ 0.5 (0.0)	4 (15.4)	
Weight loss score (%)												
- <5 - 5–10 - >10	9 (27.3)	^ 0.5 (0.0)		5 (7.4)	4 (12.5)		3 (15.8)	^ 0.5 (0.0)		3 (21.4)	3 (11.5)	
	13 (39.4)	2 (3.0)	<0.001	11 (16.2)	4 (12.5)	0.661	6 (31.6)	2 (4.9)	<0.001	1 (7.1)	6 (23.1)	0.422
	11 (33.3)	65 (97.0)		52 (76.5)	24 (75.0)		10 (52.6)	39 (95.1)		10 (71.4)	17 (65.4)	
Acute disease score												
- Zero - One - Two	24 (72.7)	22 (32.8)		33 (48.5)	13 (40.6)		13 (68.4)	7 (17.1)		9 (64.3)	17 (65.4)	
	4 (12.1)	16 (23.9)	<0.001	12 (17.6)	8 (25.0)	0.669	1 (5.3)	13 (31.7)	<0.001	1 (7.1)	5 (19.2)	0.492
	5 (15.2)	29 (43.3)		23 (33.8)	11 (34.4)		5 (26.3)	21 (51.2)		4 (28.6)	4 (15.4)	
Malnutrition risk												
- No risk - Medium - High	9 (27.3)	^ 0.5 (0.0)		5 (7.4)	4 (12.5)		3 (15.8)	^ 0.5 (0.0)		3 (21.4)	3 (11.5)	
	11 (33.3)	^ 0.5 (0.0)	<0.001	8 (11.8)	3 (9.4)	0.727	6 (31.6)	^ 0.5 (0.0)	<0.001	^ 0.5 (0.0)	5 (19.2)	0.210
	13 (39.4)	67 (100.0)		55 (80.9)	25 (78.1)		10 (52.6)	41 (100.0)		11 (78.6)	18 (6.2)	
Muscle depletion according to												
FFMI (kg/m^2^) - Normal - Depleted MUAMA (cm^2^) - Normal - Depleted												
	20 (60.6)	15 (22.4)	<0.001	25 (36.8)	10 (31.3)	0.657	8 (42.1)	7 (17.1)	0.060	7 (50.0)	13 (50.0)	1.000
	13 (39.4)	52 (77.6)		43 (63.2)	22 (68.8)		11 (57.9)	34 (82.9)		7 (50.0)	13 (50.0)	

	24 (72.7)	16 (23.9)	<0.001	30 (44.1)	10 (31.2)	0.220	4 (21.1)	15 (36.6)	0.254	10(71.4)	17 (65.4)	1.000
	9 (27.3)	51 (76.1)		38(55.9)	22 (68.8)		15 (78.9)	26 (63.4)		4 (28.6)	9 (34.6)	
Dietary intake												
- Adequate - Inadequate	24 (72.7)	16 (23.9)	<0.001	26 (38.2)	14 (43.8)	0.664	10 (52.6)	11 (26.8)	0.080	7 (50.0)	12 (46.2)	1.000
	9 (27.3)	51 (76.1)		42 (61.8)	18 (56.3)		9 (47.6)	30 (73.2)		7 (50.0)	14 (53.8)	
Healthcare scale												
- Good - Bad	27 (81.8)	43 (64.2)	0.104	47 (69.1)	23 (71.9)	0.820	12 (63.2)	29 (70.7)	0.766	11 (78.6)	18 (69.2)	0.715
	6 (18.2)	24 (35.8)		21 (30.9)	9 (28.1)		7 (36.8)	12 (29.3)		3 (21.4)	8 (30.8)	
Functional scale												
- Good - Bad	17 (51.5)	19 (28.4)	0.028	24 (35.3)	12 (37.5)	1.000	8 (42.1)	7 (17.1)	0.055	10 (71.4)	11 (42.3)	0.105
	16 (48.5)	48 (71.6)		44 (64.7)	20 (62.5)		11 (57.9)	34 (82.9)		4 28.6)	15 (57.7)	
Symptoms scale												
- Good - Bad	17 (51.5)	15 (22.4)	0.006	23 (33.8)	9 (28.1)	0.650	8 (42.1)	7 (17.1)	0.055	6 (42.9)	11 (42.3)	1.000
	16 (48.5)	52 (77.6)		45 (66.2)	23 (71.9)		11 (57.9)	34 (82.9)		8 (57.1)	15 (57.7)	

Data are presented as frequencies (n) and percentages (%). BMI: body mass index; FFMI: fat-free mass index; MUAMA: mid-upper arm muscle area. Muscle depletion was defined as FFMI < 14.6 kg/m^2^ or MUAMA < 32 cm^2^ for males and FFMI < 11.4 kg/m^2^ or MUAMA < 18 cm^2^ for females. *p* < 0.05 was considered significant using the Chi-square test or Fisher’s exact test (applied when cell counts were <5). ^ Fisher’s exact test assumes cells with 0 are approximately 0.5.

**Table 4 nutrients-16-04000-t004:** Macronutrient intake compared to the recommended dietary allowances for all cancer patients (N = 100).

Macronutrients	Intake Mean ± SD	RDA Mean ± SD	*p*-Value
Total energy (Kcal)	1574.8 ± 655.8	1796.0 ± 251.4 *	<0.001
Protein (g)	96.8 ± 36.4	119.7 ± 16.8 ^	<0.001
Carbohydrates (g)	167.0 ± 96.3	100.0 ± 0.0	<0.001
Fat (g)	57.8 ± 17.1	55.0 ± 0.0	0.108
Fiber (g)	26.9 ± 7.2	30.0 ± 0.0	<0.001
Omega 3 and 6 (g)	1.9 ± 2.3	0.25 ± 0.0	<0.001

Data are presented as means ± SD (standard deviations). Paired-sample *t*-tests were used to test the differences between the mean actual macronutrient intake and the RDA. *p*-value < 0.05 considered significant for the paired-sample *t*-test. RDA: recommended dietary allowance for cancer patients per day. * RDA was calculated for each participant as 30 Kcal/kg/day. ^ RDA was calculated for each participant as 2 g/kg/day.

**Table 5 nutrients-16-04000-t005:** Malnutrition risk, muscle depletion, quality of life scales, cancer stage, and time since diagnosis according to the dietary intake and sex of the gastrointestinal cancer patients (N = 100).

Variable	Dietary Intake			Male			Female		
	Adequate (n = 40)	Inadequate (n = 60)	*p*	Adequate (n = 21)	Inadequate (n = 39)	*p*	Adequate (n = 19)	Inadequate (n = 21)	*p*
	n (%)	n (%)		n (%)	n (%)		n (%)	n (%)	
Sex									
- Male - Female	21 (52.5)	39 (65.0)	0.149	NA	NA	NA	NA	NA	NA
	19 (47.5)	21 (35.0)							
BMI score (kg/m^2^)									
- >20 - 18.5–20 - <18.5	39 (97.5)	12 (20.0)		21 (100.0)	7 (17.9)		18 (94.7)	5 (23.8)	
	1 (2.5)	34 (56.7)	<0.001	^ 0.5 (0.0)	22 (56.4)	<0.001	1 (5.3)	12 (57.2)	<0.001
	^ 0.5 (0.0)	14 (23.3)		^ 0.5 (0.0)	10 (25.6)		^ 0.5 (0.0)	4 (19.0)	
Weight loss score (%)									
- <5 - 5–10 - >10	9 (22.5)	^ 0.5 (0.0)		3 (14.3)	^ 0.5 (0.0)		6 (31.6)	^ 0.5 (0.0)	
	15 (37.5)	^ 0.5 (0.0)	<0.001	8 (38.1)	^ 0.5 (0.0)	<0.001	7 (36.8)	^ 0.5 (0.0)	<0.001
	16 (40.0)	60 (100.0)		10 (47.6)	39 (100.0)		6 (31.6)	21 (100.0)	
Acute disease score									
- Zero - One - Two	26 (65.0)	20 (33.3)		13 (61.9)	7 (17.9)		13 (68.4)	13 (61.9)	
	8 (20.0)	12 (20.0)	0.002	5 (23.8)	9 (23.1)	0.001	3 (15.8)	3 (14.3)	0.897
	6 (15.0)	28 (46.7)		3 (14.3)	23 (59.0)		3 (15.8)	5 (23.8)	
Malnutrition risk									
- No risk - Medium risk - High risk	9 (22.5)	^ 0.5 (0.0)		3 (14.3)	^ 0.5 (0.0)		6 (31.6)	^ 0.5 (0.0)	
	11 (27.5)	^ 0.5 (0.0)	<0.001	6 (28.6)	^ 0.5 (0.0)	<0.001	5 (26.3)	^ 0.5 (0.0)	<0.001
	20 (50.0)	60 (100.0)		12 (57.1)	39 (100.0)		8 (42.1)	21 (100.0)	
Muscle depletion according to									
FFMI (kg/m^2^) - Normal - Depleted MUAMA (cm^2^) - Normal - Depleted									
	30 (75.0)	5 (8.3)	<0.001	13 (61.9)	2 (5.1)	<0.001	17 (89.5)	3 (14.3)	<0.001
	10(25.0)	55 (91.7)		8 (38.1)	37 (94.9)		2 (10.5)	18 (85.7)	

	25 (62.5)	17 (28.3)	0.0007	14 (66.7)	12 (30.8)	0.007	11 (57.9)	5 (23.8)	0.028
	15 (37.5)	43 (71.7)		7 (33.3)	27 (69.2)		8 (42.1)	16 (76.2)	
Type of cancer									
- Gastric - Colorectal	13 (32.5)	28 (46.7)	0.114	8 (38.1)	15 (38.5)	1.00	5 (26.3)	13 (61.9)	0.031
	27 (67.5)	32 (53.3)		13 (61.9)	24 (61.5)		14 (73.7)	8 (38.1)	
Healthcare scale									
- Good - Bad	32 (80.0)	38 (63.3)	0.058	17 (81.0)	24 (61.5)	0.154	15 (78.9)	14 (66.7)	0.488
	8 (20.0)	22 (36.7)		4 (19.0)	15 (38.5)		4 (21.1)	7 (33.3)	
Functional scale									
- Good - Bad	23 (57.5)	13 (21.7)	<0.001	10 (47.6)	5 (12.8)	0.005	13 (68.4)	8 (38.1)	0.067
	17 (42.5)	47 (78.3)		11 (52.4)	34 (87.2)		6 (31.6)	13 (61.9)	
Symptoms scale									
- Good - Bad	26 (65.0)	6 (10.0)	<0.001	13 (61.9)	2 (5.1)	<0.001	13 (68.4)	4 (19.0)	0.003
	14 (35.0)	54 (90.0)		8 (38.1)	37 (94.9)		6 (31.6)	17 (81.0)	
Cancer stage									
- I and II - III and IV	24 (60.0)	9 (15.0)	<0.001	10 (47.6)	9 (23.1)	0.080	7 (36.8)	7 (33.3)	1.000
	16 (40.0)	51 (85.0)		11 (52.4)	30 (76.9)		12 (63.2)	14 (66.7)	
Time since diagnosis									
- ≤4 months - >4 months	26 (65.0)	42 (70.0)	0.664	10 (47.6)	25 (64.1)	0.276	16 (84.2)	17 (81.0)	1.000
	14 (35.0)	18 (30.0)		11 (52.4)	14 (35.9)		3 (15.8)	4 (19.0)	

Data are presented as frequencies (n) and percentages (%). BMI: body mass index; FFMI: fat-free mass index; MUAMA: mid-upper arm muscle area. Muscle depletion was defined as FFMI < 14.6 kg/m^2^ or MUAMA < 32 cm^2^ for males and FFMI < 11.4 kg/m^2^ or MUAMA < 18 cm^2^ for females. *p* < 0.05 was considered significant using the Chi-square test or Fisher’s exact test (applied when cell counts were <5). ^ Fisher’s exact test assumes cells with 0 are approximately 0.5.

**Table 6 nutrients-16-04000-t006:** Summary of the mediation analysis results.

Total Effect: Dietary Intake -> Malnutrition Risk	Direct Effect: Dietary Intake -> Malnutrition Risk	Relationship	Indirect Effect	95% Confidence Interval		t-Statistics	Conclusions
				Lower	Upper		
0.4012 (0.0001)	0.2849 (0.0091)	Dietary intake -> FFMI -> Malnutrition risk	0.0285	0.0416	0.1556	0.5852	Partial mediation
		Dietary intake -> MUAMA -> Malnutrition risk	0.0878	0.0145	0.2105	1.5791	Partial mediation

FFMI: fat-free mass index; MUAMA: mid-upper arm muscle area. *p* < 0.05 is significant for mediation analysis using Hayes Process Macro Model 4 with multiple mediators and covariates.

**Table 7 nutrients-16-04000-t007:** Binary logistic regression model ^1^ for the malnutrition risk outcome.

Variables in the Equation	B	S.E.	*p*-Value	OR	95% C.I. for OR	
					Lower	Upper
Sex	2.676	1.260	0.034	14.526	1.229	171.701
Energy intake (Kcal/day)	−0.007	0.003	0.020	0.993	0.987	0.999
MUAMA (cm^2^)	2.774	1.143	0.015	16.024	1.706	150.507
Type of cancer	2.667	1.126	0.018	14.393	1.583	130.867

^1^ Hosmer and Lemeshow test: χ^2^ = 2.237; *df* = 8; *p*-value = 0.973. B: regression coefficient; S.E.: standard error of the regression coefficient; C.I.: confidence interval; OR: odds ratio; MUAMA: mid-upper arm muscle area. *p*-value ≤ 0.05 is statistically significant.

## Data Availability

Data will be made available upon special request to the corresponding author due to the inclusion of sensitive and personal information that cannot be disclosed for ethical reasons.

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
