# Peer review of "Gender-Specific Malnutrition and Muscle Depletion in Gastric and Colorectal Cancer: Role of Dietary Intake in a Jordanian Cohort"

_nutrients, 2024, doi:10.3390/nu16234000_

Round 1
Reviewer 1 Report
Comments and Suggestions for Authors
Thanks for the chance to review this study based upon nutritional assessments carried out on a convenience sample of 100 UGI and LGI cancer patients at a single tertiary centre in Jordan. The study sought to elucidate between-sex and between-cancer differences, but a convenience sample at a tertiary hospital, uncontrolled for cancer stage (and in fact no data is presented on time since diagnosis, cancer stage, prior treatment or reason for clinic attendance) means that patient disease state and behaviour in attending the clinic (which is known to differ between men and women) is likely to be a major uncontrolled confounder.
Without such information, it is impossible to draw any inference about why the observed between-sex and between-cancer differences were seen. In turn, while the lack of cancer stage is mentioned as a limitation, the reach questions cannot be answered with any confidence - were men in clinic more likely to have advanced stage disease? did gastric cancer patients have earlier or later surgery for their diagnosis than colorectal cancer patients (and thereby limit study inclusion in one or other groups)?
Overall, without data about tumour stage and time since diagnosis (which should be readily obtainable from clinical records) I feel the study design is flawed and cannot answer its research questions adequately.
Author Response
Comments: Thanks for the chance to review this study based upon nutritional assessments carried out on a convenience sample of 100 UGI and LGI cancer patients at a single tertiary centre in Jordan. The study sought to elucidate between-sex and between-cancer differences, but a convenience sample at a tertiary hospital, uncontrolled for cancer stage (and in fact no data is presented on time since diagnosis, cancer stage, prior treatment or reason for clinic attendance) means that patient disease state and behaviour in attending the clinic (which is known to differ between men and women) is likely to be a major uncontrolled confounder.
Without such information, it is impossible to draw any inference about why the observed between-sex and between-cancer differences were seen. In turn, while the lack of cancer stage is mentioned as a limitation, the reach questions cannot be answered with any confidence - were men in clinic more likely to have advanced stage disease? did gastric cancer patients have earlier or later surgery for their diagnosis than colorectal cancer patients (and thereby limit study inclusion in one or other groups)?
Overall, without data about tumour stage and time since diagnosis (which should be readily obtainable from clinical records) I feel the study design is flawed and cannot answer its research questions adequately.
Response: Thank you for your valuable feedback and for taking the time to review our manuscript. We contacted the IT office at the university hospital, which provided the necessary data from the patients' medical records. After obtaining information on patients’ cancer stages (determined using the TNM classification) during the period of data collection and the time since diagnosis, we incorporated these variables into our analysis to assess their associations with malnutrition risk and muscle depletion. These variables were categorized into two groups, as described in the statistical analysis section, and the results are presented in Table 3.
Additionally, we analyzed the associations between gender and cancer stage with malnutrition risk, muscle depletion, dietary intake, and quality of life scales. While the time since diagnosis did not show a significant association with malnutrition risk in the study group, cancer stage demonstrated a statistically significant relationship with malnutrition risk, particularly in advanced stages (III and IV).
These new findings are included in a dedicated section of the results (Section 3.3) and were also discussed in the discussion section and summarized in the abstract. Furthermore, we included the cancer stage as a covariate in the mediation analysis model and the results were modified accordingly (section 3.5). Also, the reasons for attending clinics have been clarified in Section 2.1 of the methods. Finally, we removed the absence of cancer stage data from the study limitations, as this issue has now been addressed.
We appreciate your thoughtful suggestions, which have strengthened the manuscript.
Reviewer 2 Report
Comments and Suggestions for Authors
1. Authors should emphasize how many patients in the study had gastric cancer and how many had colorectal cancer. Also, authors should specify the type of gastric cancer (diffuse or intestinal) as each subtype impacts malnutrition risk differently.
2. In the study authors assesses malnutrition risk using anthropometric measurements and tools like MUST, FFMI, and MUAMA but does not include some other biochemical markers such as serum albumin. As is known, serum albumin is often used in clinical settings as a reliable marker for malnutrition and can provide additional depth to the malnutrition assessment. Therefore, I believe that the authors should include the value of albumin in their article in assessing malnutrition.
3. The study does not account for cancer staging, which could significantly impact malnutrition risk and muscle depletion rates. Malnutrition and cachexia are often more severe in advanced stages of cancer. It is very important that the authors include cancer stage as a variable because this would allow them to much better contextualize the relationship between malnutrition and disease progression.
4. The text below each table needs to be edited because the way it is written now looks cluttered.
Author Response
Comment1: Authors should emphasize how many patients in the study had gastric cancer and how many had colorectal cancer. Also, authors should specify the type of gastric cancer (diffuse or intestinal) as each subtype impacts malnutrition risk differently.
Response: Thank you for your valuable feedback. As noted, we have already specified the number of gastric (n = 41) and colorectal (n = 59) cancer patients in Table 1 of the manuscript. Regarding the specific subtypes of gastric cancer, our sample comprises predominantly the intestinal subtype of adenocarcinoma (n = 40), with only one case of the diffuse subtype (n = 1). Given this distribution, we opted to group both subtypes under the broader category of gastric cancer in our analyses. This approach reflects the sample’s composition and allows for a more streamlined analysis without compromising interpretative clarity. Nonetheless, we acknowledge that each subtype may have distinct implications for malnutrition risk.
Comment 2: In the study authors assesses malnutrition risk using anthropometric measurements and tools like MUST, FFMI, and MUAMA but does not include some other biochemical markers such as serum albumin. As is known, serum albumin is often used in clinical settings as a reliable marker for malnutrition and can provide additional depth to the malnutrition assessment. Therefore, I believe that the authors should include the value of albumin in their article in assessing malnutrition.
Response: Thank you for your insightful comment. We agree that serum albumin is a reliable biochemical marker for malnutrition and is widely used in clinical settings, particularly when anthropometric measurements and malnutrition assessment tools are unavailable or when confirmation of malnutrition is required. Serum albumin provides valuable information on nutritional status and correlates with muscle mass, making it useful for indirect assessment of body composition changes, such as sarcopenia. This can be particularly relevant for cancer patients who are at high risk for muscle wasting and cachexia. Including serum albumin in malnutrition assessment could indeed add depth to our findings. However, due to limitations in our data collection protocol, we were unable to obtain blood samples from patients to measure this biomarker. We have acknowledged this as one of the study's limitations, and we appreciate your suggestion to include serum albumin in future studies to enhance the assessment of malnutrition.
Comment 3: The study does not account for cancer staging, which could significantly impact malnutrition risk and muscle depletion rates. Malnutrition and cachexia are often more severe in advanced stages of cancer. It is very important that the authors include cancer stage as a variable because this would allow them to much better contextualize the relationship between malnutrition and disease progression.
Response: Cancer staging was obtained from the medical records of the participating patients and was incorporated into the data analysis, as detailed in Table 3 and Section 3.3 and 3.5 of the Results.
Comment 4: The text below each table needs to be edited because the way it is written now looks cluttered.
Response: The text below the tables has been revised and formatted to ensure clarity and a clean, uncluttered appearance.